# Generalizable brain network markers of major depressive disorder across multiple imaging sites

**Ayumu Yamashita**[1]*, **Yuki Sakai**[1], **Takashi Yamada**[1,2], **Noriaki Yahata**[1,3,4,5], **Akira Kunimatsu**[6,7], **Naohiro Okada**[3,8], **Takashi Itahashi**[2], **Ryuichiro Hashimoto**[1,2,9], **Hiroto Mizuta**[10], **Naho Ichikawa**[11], **Masahiro Takamura**[11], **Go Okada**[11], **Hirotaka Yamagata**[12], **Kenichiro Harada**[12], **Koji Matsuo**[12,13], **Saori C. Tanaka**[1], **Mitsuo Kawato**[1,14]*, **Kiyoto Kasai**[1,3,8], **Nobumasa Kato**[1,2], **Hidehiko Takahashi**[10,15], **Yasumasa Okamoto**[11], **Okito Yamashita**[1,14], **Hiroshi Imamizu**[1,16]

1 Brain Information Communication Research Laboratory Group, Advanced Telecommunications Research Institutes International, Kyoto, Japan, 2 Medical Institute of Developmental Disabilities Research, Showa University, Tokyo, Japan, 3 Department of Neuropsychiatry, Graduate School of Medicine, The University of Tokyo, Tokyo, Japan, 4 Institute for Quantum Life Science, National Institutes for Quantum and Radiological Science and Technology, Chiba, Japan, 5 Department of Molecular Imaging and Theranostics, National Institute of Radiological Sciences, National Institutes for Quantum and Radiological Science and Technology, Chiba, Japan, 6 Department of Radiology, IMSUT Hospital, Institute of Medical Science, The University of Tokyo, Tokyo, Japan, 7 Department of Radiology, Graduate School of Medicine, The University of Tokyo, Tokyo, Japan, 8 The International Research Center for Neurointelligence (WPI-IRCN) at the University of Tokyo Institutes for Advanced Study (UTIAS), Tokyo, Japan, 9 Department of Language Sciences, Tokyo Metropolitan University, Tokyo, Japan, 10 Department of Psychiatry, Kyoto University Graduate School of Medicine, Kyoto, Japan, 11 Department of Psychiatry and Neurosciences, Hiroshima University, Hiroshima, Japan, 12 Division of Neuropsychiatry, Department of Neuroscience, Yamaguchi University Graduate School of Medicine, Yamaguchi, Japan, 13 Department of Psychiatry, Faculty of Medicine, Saitama Medical University, Saitama, Japan, 14 Center for Advanced Intelligence Project, RIKEN, Tokyo, Japan, 15 Department of Psychiatry and Behavioral Sciences, Graduate School of Medical and Dental Sciences, Tokyo Medical and Dental University, Tokyo, Japan, 16 Department of Psychology, Graduate School of Humanities and Sociology, The University of Tokyo, Tokyo, Japan

* ayumu@atr.jp (AY); kawato@atr.jp (MK)

**Data Availability Statement:** Relevant data which used in the figures are within its Supporting Information files. The raw data utilized in this study can be downloaded publicly from the DecNef

## Abstract

Many studies have highlighted the difficulty inherent to the clinical application of fundamental neuroscience knowledge based on machine learning techniques. It is difficult to generalize machine learning brain markers to the data acquired from independent imaging sites, mainly due to large site differences in functional magnetic resonance imaging. We address the difficulty of finding a generalizable marker of major depressive disorder (MDD) that would distinguish patients from healthy controls based on resting-state functional connectivity patterns. For the discovery dataset with 713 participants from 4 imaging sites, we removed site differences using our recently developed harmonization method and developed a machine learning MDD classifier. The classifier achieved an approximately 70% generalization accuracy for an independent validation dataset with 521 participants from 5 different imaging sites. The successful generalization to a perfectly independent dataset acquired from multiple imaging sites is novel and ensures scientific reproducibility and clinical applicability.

Project Brain Data Repository at https://bicr-resource.atr.jp/srpbsopen/ and https://bicr.atr.jp/dcn/en/download/harmonization/.

**Funding:** This study was conducted under the contract research "Brain/MINDS Beyond" Grant Number JP18dm0307008, supported by the Japan Agency for Medical Research and Development (AMED) while using data obtained from the database project supported by "Development of BMI Technologies for Clinical Application" of the Strategic Research Program for Brain Sciences JP17dm0107044 (AMED). This study was also supported by Grant Number JP18dm0307002, JP18dm0307004, and JP19dm0307009 (AMED). M.K., H.I. and A.Y. were partially supported by the ImPACT Program of the Council for Science, Technology and Innovation (Cabinet Office, Government of Japan). K.K. was partially supported by the International Research Center for Neurointelligence (WPI-IRCN) at The University of Tokyo Institutes for Advanced Study (UTIAS) and JSPS KAKENHI 16H06280 (Advanced Bioimaging Support). H.I. was partially supported by JSPS KAKENHI 18H01098, 18H05302, and 19H05725. The funders had no role in study design, data collection and analysis, decision to publish, or preparation of the manuscript.

**Competing interests:** I have read the journal's policy and the authors of this manuscript have the following competing interests: M.K., N.Y., R.H., H.I., N.K. and K.K are inventors of a patent owned by Advanced Telecommunications Research (ATR) Institute International related to the present work [PCT/JP2014/061543 (WO2014178322)]. M.K., N.Y., R.H., N.K. and K.K. are inventors of a patent owned by ATR Institute International related to the present work [PCT/JP2014/061544 (WO2014178323)]. M.K. and N.Y. are inventors of a patent application submitted by ATR Institute International related to the present work [JP2015-228970]. A.Y. and M.K. are inventors of a patent application submitted by ATR Institute International related to the present work [JP2018-192842].

**Abbreviations:** AAL, anatomical automatic labeling; ASD, autism spectrum disorder; AUC, area under the curve; BDI-II, Beck Depression Inventory-II; BOLD, blood-oxygen-level–dependent; CI, confidence interval; CREST, Core Research for Evolutional Science and Technology; CSF, cerebrospinal fluid; DSM, Diagnostic and Statistical Manual of Mental Disorders; FC, functional connectivity; FD, frame-wise displacement; HC, healthy control; HCP, Human Connectome Project; HKH, Hiroshima Kajikawa Hospital; HRC, Hiroshima Rehabilitation Center; HUH, Hiroshima University Hospital; LASSO, least absolute

## Introduction

Major depressive disorder (MDD) is a highly heterogeneous psychiatric disorder, and a lumping approach that treats MDD as a single entity has been criticized since early 2000 [1,2]. Such heterogeneity in psychiatric disorders has motivated the Research Domain Criteria (RDoC) initiative, which aims to redefine and identify subtypes of psychiatric disorders in terms of biological systems, without relying on a diagnosis based solely on symptoms and signs [3]. This initiative is expected to inform our understanding of heterogeneous and overlapping clinical presentations of psychotic disorders [4–7]. In particular, resting-state functional magnetic resonance imaging (rs-fMRI) is a useful modality to this end because it enables us to noninvasively investigate whole brain functional connectivity (FC) in diverse patient populations [8,9]. rs-fMRI allows for the quantification of the FC of correlated, spontaneous blood-oxygen-level–dependent (BOLD) signal fluctuations [10]. Machine learning algorithms have emerged as powerful tools for the analysis of a large number of FCs (usually between 10,000 and 100,000 for an individual). According to the original idea of the RDoC initiative, subtyping and redefining psychiatric disorders should be based on solely biological and cognitive measurements (FCs in the current context) without relying on traditional symptom-based categories [11–13]. An unsupervised learning technique is the first candidate for this data-driven approach. However, the number of explanatory variables, FCs, is huge (10,000 to 100,000), while the sample size, i.e., the number of participants, usually ranges between 100 and 1,000. Thus, overfitting to noise in the data by machine learning algorithms and the resultant inflation of prediction performance can easily occur [12,14,15]. This situation makes it difficult to directly apply an unsupervised learning algorithm to FC data.

To address this problem and successfully subtype and redefine psychiatric disorders, we proposed the following hierarchical supervised/unsupervised approach, which was shown to have been partially successful in several studies [16–18]. First, we identified a relatively small number of FCs that reliably distinguish healthy controls (HCs) and patients with psychiatric disorders using a supervised learning algorithm. Our purpose at this stage is to find potentially relevant biological dimensions to psychiatric disorders. Thus, lumping categories such as "MDD" as a single entity may provide useful information with supervised learning to search for relevant biological dimensions. We can use the identified FCs not only for a brain network biomarker of the psychiatric disorder but also for biologically meaningful dimensions related to the disorder. Second, we applied unsupervised learning to these low biological dimensions to redefine and find subtypes of psychiatric disorders. For instance, we were able to achieve subtyping of MDD by locating patients with MDD in these dimensions [19]. Furthermore, locating different psychiatric disorders (e.g., MDD, schizophrenia [SCZ], and autism spectrum disorder [ASD]) in these dimensions may reveal the relationships among the disorders (multi-disorder spectrum) [16–18]. As such, although our approach starts with supervised learning based on a lumping category such as a diagnosis, such category is only used as a single piece of information to recruit relevant FCs to psychiatric disorders. Our final goal is to understand psychiatric disorders in the biological dimensions while avoiding overfitting to noise in the discovery dataset and ensuring generalization performance for the independent data in completely different multiple imaging sites.

Whether a brain network marker constructed in the first stage generalizes to the data acquired from multiple completely different imaging sites is a crucial issue for the above hierarchical supervised/unsupervised approach [20–22]. However, an increasing number of studies have highlighted the difficulty in generalization of the brain network marker to the data acquired from multiple completely independent imaging sites, even using the supervised learning method [14,23]. For example, in a recent paper by Drysdale, which is one of the most

shrinkage and selection operator; MCC, Matthews correlation coefficient; MDD, major depressive disorder; MNI, Montreal Neurological Institute; NPV, negative predictive value; PPV, positive predictive value; RDoC, Research Domain Criteria; ROI, region of interest; rs-fMRI, resting-state functional magnetic resonance imaging; rTMS, repetitive transcranial magnetic stimulation; SCID, Structured Clinical Interview for DSM; SCZ, schizophrenia; SRPBS, Strategic Research Program for Brain Science; SVM, support vector machine; UYA, Yamaguchi University; WM, white matter.

successful brain network markers of MDD, the classification accuracy for MDD in completely independent imaging sites was 68.8% for 16 patients from 1 site, which represents only 3% of the validation cohort ([12]).

Here, we targeted MDD, the world's most serious psychiatric disorder in terms of its social repercussions [24,25]. To achieve the final goal (redefinition and subtyping), it is an absolutely necessary prerequisite to first treat MDD as a single entity using diagnoses as supervised data and create the brain network marker that generalizes to a completely independent data collected from multiple imaging sites. We considered and satisfied 3 issues and conditions to ensure generalization of our network marker of MDD to the independent validation dataset, which does not include imaging sites of the discovery dataset. First, we used our recently developed harmonization method, which could reduce site differences in FC [26]. According to our recent study, the differences in resting-state FCs for different imaging sites consist of measurement bias due to differences in fMRI protocols and MR scanners and sampling bias due to recruitment of different participant populations. The magnitude of the measurement bias was larger than the effects of disorders including MDD, and the magnitude of the sampling bias was comparable with the effects of disorders [26]. Therefore, a reduction in the site difference in FC is essential for the generalization of network models in the validation dataset. Second, we validated our network marker using a perfectly independent and large cohort collected from multiple completely different imaging sites. Specifically, for constructing a brain network marker, we used an rs-fMRI discovery dataset with 713 participants (149 patients with MDD) from 4 imaging sites (Center of Innovation in Hiroshima University (COI), Kyoto University (KUT), Showa University (SWA), and University of Tokyo (UTO)). This discovery dataset was collected by a Japanese nationwide database project called the Strategic Research Program for Brain Science (SRPBS, https://bicr.atr.jp/decnefpro/) since 2014. After constructing the brain network marker, we examined generalization of the network marker to an independent validation dataset with 449 participants (185 patients with MDD) from 4 different imaging sites (Hiroshima Kajikawa Hospital [HKH], Hiroshima Rehabilitation Center [HRC], Hiroshima University Hospital [HUH], and Yamaguchi University [UYA]). This validation dataset was formed after the construction of the network marker and was acquired for other projects since 2008 independently of the SRPBS. We further assessed the generalization performance to the data collected from a country other than Japan by using publicly available rs-fMRI dataset with 72 participants (51 patients with MDD) from OpenNeuro (https://openneuro.org/datasets/ds002748/versions/1.0.0). Furthermore, we used another dataset of 75 HCs, 154 patients with SCZ, and 121 patients with ASD to investigate the multi-disorder spectrum. In total, we used 1,584 participants' data in this study. Furthermore, unlike previous studies that restricted the subtype of MDD [12,16], we targeted all patients with MDD without restricting according to the depression subtype in order to enable future subtyping in the biological dimensions, which is beyond the purpose of the current paper. Third, we carefully avoided overfitting noise in the discovery dataset. As explained above, the number of explanatory variables is typically larger than the sample size in the rs-fMRI study; thus, overfitting to noise in the discovery dataset by machine learning algorithms and resultant inflation of prediction performance can happen easily unless special precautions are taken. We used a sparse machine learning algorithm with the least absolute shrinkage and selection operator (LASSO) to avoid overfitting to noise and selected only essential FCs [27]. As a result, for the first time, to our knowledge, we developed a generalizable brain network marker for MDD without restricting to certain subtypes such as treatment-resistant or melancholic MDD.

Previous studies have shown that a diagnosis of MDD based on the Diagnostic and Statistical Manual of Mental Disorders (DSM) has a low inter-rater agreement (kappa = 0.28) [28,29].

In this study, we have developed a brain network marker that can objectively predict a diagnosis from structured interviews, which have relatively high inter-rater agreement, such as the Structured Clinical Interview for DSM (SCID) (kappa = 0.64~0.74) [30,31] or the Mini-International Neuropsychiatric Interview (high agreement with SCID: kappa = 0.85) [32]. To objectively compare and verify the stability of the diagnoses based on DSM and brain FC, we performed a simulation. Our simulation will show that our brain network marker of MDD may more objectively and stably diagnose MDD than a diagnosis based on DSM even if we considered the variance across multiple scans, different fMRI scanners, and different imaging sites. Of note, although previous studies suggest that it is especially difficult to make a diagnosis that differentiates MDD from bipolar disorder [33], it is relatively straightforward for an experienced clinician to distinguish between MDD and healthy status in clinical practice. Therefore, the utility of the rs-fMRI-based brain network marker is to understand underlying pathophysiological mechanisms of the illness and to guide treatment choice and the future development of novel interventions for a given disorder.

## Results

### Datasets

We used 2 rs-fMRI datasets for the analyses. The "discovery dataset" contained data from 713 participants (564 HCs from 4 sites, 149 patients with MDD from 3 sites; Table 1), and the "independent validation dataset" contained data from 521 participants (285 HCs and 236 patients with MDD from 5 independent sites; Table 1). Most data utilized in this study can be downloaded publicly from the DecNef Project Brain Data Repository (https://bicr-resource. atr.jp/srpbsopen/ and https://bicr.atr.jp/dcn/en/download/harmonization/) and OpenNeuro (https://openneuro.org/datasets/ds002748/versions/1.0.0). The imaging protocols and data availability statement of each site is described in S1 Table. Depression symptoms were evaluated using the Beck Depression Inventory-II (BDI-II) score obtained from most participants in each dataset. Clinical details such as medication information and the presence of comorbidities in patients with MDD are described in S2 Table.

### Site difference control in FC

Classical preprocessing was performed, and FC was defined based on a functional brain atlas consisting of 379 nodes (regions) covering the whole brain [34]. The Fisher's $z$-transformed Pearson correlation coefficients between the preprocessed BOLD signal time courses of each possible pair of nodes were calculated and used to construct $379 \times 379$ symmetrical connectivity matrices in which each element represents a connection strength, or edge, between 2 nodes. We used 71,631 connectivity values ($379 \times 378/2$) of the lower triangular matrix of the connectivity matrix. To control for site differences in the FC, we applied a traveling subject harmonization method to the discovery dataset [26]. In this method, the measurement bias (the influence of the difference in the properties of MRI scanners, such as the imaging parameters, field strength, MRI manufacturer, and scanner model) was estimated by fitting the regression model to the FC values of all participants from the discovery dataset and the traveling subject dataset, wherein multiple participants travel to multiple sites to assess measurement bias (see Control of site differences in the Materials and methods section). This method enabled us to subtract only the measurement bias while leaving important information due to differences in subjects among imaging sites. We applied the ComBat harmonization method [35–38] to control for site differences in the FC of the independent validation dataset because we did not have a traveling subject dataset for those sites.

**Table 1. Demographic characteristics of participants in both datasets.**

| Site | HC | | | | MDD | | | | All | | | |
|---|---|---|---|---|---|---|---|---|---|---|---|---|
| | Number | Male/Female | Age (y) | BDI | Number | Male/Female | Age (y) | BDI | Number | Male/Female | Age (y) | BDI |
| **Discovery dataset** | | | | | | | | | | | | |
| Center of Innovation in Hiroshima University (COI) | 124 | 46/78 | 51.9 ± 13.4 | 8.2 ± 6.3 | 70 | 31/39 | 45.0 ± 12.5 | 26.2 ± 9.9 | 194 | 77/117 | 49.4 ± 13.5 | 14.7 ± 11.7 |
| Kyoto University (KUT) | 169 | 100/69 | 35.9 ± 13.6 | 6.0 ± 5.4 | 17 | 11/6 | 43.9 ± 13.3 | 27.7 ± 10.1 | 186 | 111/75 | 36.7 ± 13.7 | 8.3 ± 9.1 |
| Showa University (SWA) | 101 | 86/15 | 28.4 ± 7.9 | 4.4 ± 3.8 | 0 | - | - | - | 101 | 86/15 | 28.4 ± 7.9 | 4.4 ± 3.8 |
| University of Tokyo (UTO) | 170 | 78/92 | 35.6 ± 17.5 | 6.7 ± 6.5 | 62 | 36/26 | 38.7± 11.6 | 20.4 ± 11.4 | 232 | 114/118 | 36.4 ± 16.2 | 14.5 ± 11.8 |
| Summary | 564 | 310/254 | 38.0 ± 16.1 | 6.3 ± 5.6 | 149 | 78/71 | 42.3 ± 12.5 | 24.9 ± 10.7 | 713 | 388/325 | 38.9 ±15.5 | 10.7 ± 10.6 |
| **Independent validation dataset** | | | | | | | | | | | | |
| Hiroshima Kajikawa Hospital (HKH) | 29 | 12/17 | 45.4 ± 9.5 | 5.1 ± 4.6 | 33 | 20/13 | 44.8 ± 11.5 | 28.5 ± 8.7 | 62 | 32/30 | 45.1 ± 10.5 | 17.6 ± 13.7 |
| Hiroshima Rehabilitation Center (HRC) | 49 | 13/36 | 41.7 ± 11.7 | 9.1 ± 8.5 | 16 | 6/10 | 40.5 ± 11.5 | 35.3 ± 9.5 | 65 | 19/46 | 41.4 ± 11.5 | 15.6 ± 14.3 |
| Hiroshima University Hospital (HUH) | 66 | 29/37 | 34.6 ± 13.0 | 6.9 ± 5.9 | 57 | 32/25 | 43.3 ± 12.2 | 30.9 ± 9.0 | 123 | 61/62 | 38.6 ± 13.3 | 18.0 ± 14.1 |
| Yamaguchi University (UYA) | 120 | 50/70 | 45.9 ± 19.5 | 7.1 ± 5.6 | 79 | 36/43 | 50.3 ± 13.6 | 29.7 ± 10.7 | 199 | 86/113 | 47.6 ± 17.5 | 16.0 ± 13.6 |
| Summary | 264 | 104/160 | 42.2 ± 16.5 | 7.2 ± 6.3 | 185 | 94/91 | 46.3 ± 13.0 | 30.3 ± 9.9 | 449 | 198/251 | 43.9 ± 15.3 | 16.7 ± 13.9 |
| **OpenNeuro dataset** | | | | | | | | | | | | |
| OTHER | 21 | 6/15 | 33.8±8.49 | - | 51 | 13/38 | 32.7±8.94 | - | 72 | 19/53 | 33.1±8.73 | - |

Demographic distribution of sex rate was matched between the MDD and HC populations in the discovery dataset ($P > 0.05$) but not for age or BDI ($P < 0.05$). All demographic distributions were not matched between the MDD and HC populations in the independent validation dataset ($P < 0.05$). The link to OpenNeuro dataset is https://openneuro.org/datasets/ds002748/versions/1.0.0

BDI, Beck Depression Inventory-II; HC, healthy control; MDD, major depressive disorder.

## Reproducible FCs related to MDD diagnosis

Using a simple mass univariate analysis, we first investigated the reproducibility of the effect size by diagnosis on individual FC across the discovery and validation datasets. For the effect of the diagnosis on each FC, we calculated the difference in the FC value between the HCs and the MDDs (*t*-value). Fig 1 shows the diagnosis effect size for the discovery dataset in the abscissa and that for the validation dataset in the ordinate for each FC. To statistically evaluate the reproducibility of the effect on FCs, we calculated the Pearson's correlation of the effect sizes (*t*-values) between the discovery and validation datasets. We compared this Pearson's correlation value with the distributions of the Pearson's correlation in the shuffled data in which diagnosis labels were permuted across subjects (permutation test). We found a significant correlation between the 2 datasets ($r_{(71,631)} = 0.494$, 95% confidence interval (CI) = [0.488 0.499], $R^2 = 0.24$, [permutation test, $P < 0.001$, 1-sided]). This result indicates that resting-state FCs contain consistent information regarding MDD diagnosis across the 2 datasets, even if the 2 datasets were acquired from completely different sites.

**Fig 1. Results of mass univariate analysis.** Reproducibility across the 2 datasets regarding diagnosis effects. Scatter plot and histograms of the diagnosis effect size (the difference in mean functional connectivity strengths between patients with depression and healthy groups: *t*-value). Each point in the scatter plot represents the diagnosis effect in the discovery dataset in the abscissa and that for the independent validation dataset in the ordinate for each functional connectivity. The original data is in black, while the shuffled data in which subject information was permuted is in gray. The numerical data used in this figure are included in S1 Data.

## Brain network marker of MDD diagnosis generalized to MDD data obtained from completely different multi-sites

We constructed a brain network marker for MDD, which distinguished between HCs and MDD patients, using the discovery dataset based on 71,631 FC values. Based on our previous studies [16–18,39], we assumed that psychiatric disorder factors were not associated with whole brain connectivity, but rather with a specific subset of connections. Therefore, we used logistic regression with LASSO, a sparse machine learning algorithm, to select the optimal subset of FCs [40]. We have already succeeded in constructing generalizable brain network markers of ASD, melancholic MDD, SCZ, and obsessive compulsive disorder [16–18,39] by using a similar sparse estimation method that automatically selects the most important connections. We also tried a support vector machine (SVM) for classification instead of LASSO. However, the performance was not improved compared with that with LASSO (S1 Text).

To estimate the weights of logistic regression and a hyperparameter that determines how many FCs were used, we conducted a nested cross-validation procedure (Fig 2) (see Constructing the MDD classifier using the discovery dataset in the Materials and methods section). We first divided the whole discovery dataset into the training set (9 folds out of 10 folds), which was used for training a model, and the test set (1 fold out of 10 folds), which was used for testing the model. To avoid bias due to the difference in the number of patients with MDD

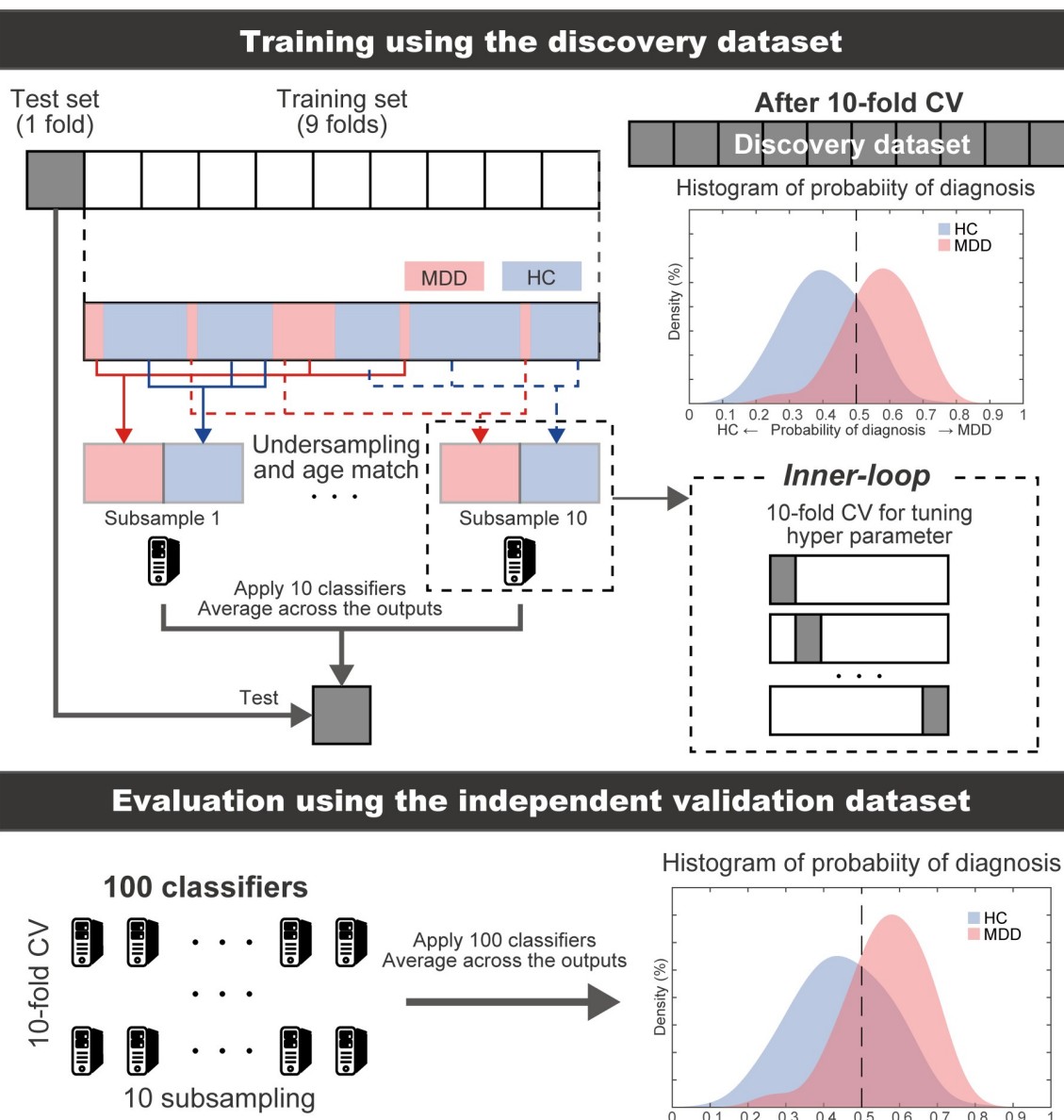

**Fig 2. Schematic representation of the procedure for training the MDD classifier and evaluation of predictive power.** The MDD classifier was constructed using a nested cross-validation procedure in the discovery dataset. We also used undersampling and subsampling techniques. Generalization performances were evaluated by applying the constructed classifiers to the independent validation dataset. The machine learning classifiers are represented by PC cartoons. CV, cross-validation; HC, healthy control; MDD, major depressive disorder.

and HCs, we used an undersampling method for equalizing the numbers between the MDD and HC groups [41]. Since only a subset of training data is used after undersampling, we repeated the random sampling procedure 10 times (i.e., subsampling). When we performed undersampling and subsampling procedures, we matched the mean age between MDD and HC groups in each subsample. We then fitted a model to each subsample while tuning a regularization parameter in an inner loop of nested cross validation, resulting in 10 classifiers. The mean classifier-output value (diagnostic probability) was considered indicative of the classifier output. Diagnostic probability values of >0.5 were considered to be indicative of an MDD

diagnosis. We calculated the area under the curve (AUC), accuracy, sensitivity, specificity, positive predictive value (PPV), and negative predictive value (NPV). Furthermore, we evaluated the classifier performance for the unbalanced dataset using the Matthews correlation coefficient (MCC) [42,43], which takes into account the ratio of the confusion matrix size.

The classifier distinguished MDD and HC populations with an accuracy of 66% in the discovery dataset. The corresponding AUC was 0.74, indicating acceptable discriminatory ability. Fig 3A shows that the 2 diagnostic probability distributions of the MDD and HC populations were clearly separated by the 0.5 threshold (right, MDD; left, HC) for the discovery dataset. The sensitivity, specificity, PPV, and NPV were 72%, 65%, 0.34, and 0.90, respectively. This classifier led to an acceptable MCC of 0.30. We found that acceptable classification accuracy was achieved for the full dataset as well as for the individual datasets from 3 of the imaging sites (Fig 3B) to similar degrees. Only HC individuals were identified in the SWA dataset; however, notably, its probability distribution was comparable to the HC populations at other sites.

We tested the generalizability of the classifier using an independent validation dataset. We created 100 classifiers of MDD (10-fold × 10 subsamples); therefore, we applied all trained classifiers to the independent validation dataset. Next, we averaged the 100 outputs (diagnostic probability) for each participant and considered the participant as a patient with MDD if the averaged diagnostic probability value was >0.5. The classifier distinguished the MDD and HC populations with an accuracy of 66% in the independent validation dataset. If the accuracy for the validation dataset is much smaller than that of the discovery dataset, overfitting is strongly suggested, and the reproducibility of the results is put into doubt. In our case, 66% accuracy for the validation dataset was actually same with 66% accuracy for the discovery dataset, and this concern does not apply. The corresponding AUC was 0.74 (permutation test, $P < 0.01$, 1-sided), indicating an acceptable discriminatory ability. Fig 3C shows that the 2 diagnostic probability distributions of the MDD and HC populations were clearly separated by the 0.5 threshold (right, MDD; left, HC). The sensitivity, specificity, PPV, and NPV were 72%, 61%, 0.60, and 0.73, respectively. This approach led to an acceptable MCC of 0.33 (permutation test, $P < 0.01$, 1-sided). In addition, acceptable classification accuracy was achieved for the individual datasets of the 5 imaging sites (Fig 3D).

To investigate whether our classifier can be generalized to milder depression, we applied our classifier to patients with MDD with low BDI scores (score $\leq 20$, $n = 30$) in the independent validation dataset. As a result, 21 of the 30 patients were correctly classified as having MDD (accuracy of 70%), a similar performance level to when the classifier was applied to all patients with MDD. In addition, we assessed whether the proportion of antidepressants, anxiolytics, antipsychotics, and mood stabilizers used in the milder group and severe group were statistically different. We found that the milder group used significantly lesser antidepressants than the severe group (antidepressant: milder = 19/30, severe = 124/155, $z = 2.00$, $P = 0.046$; anxiolytic: milder = 10/30, severe = 65/155, $z = 0.88$, $P > 0.38$; antipsychotic: milder = 9/30, severe = 33/155, $z = 1.04$, $P > 0.3$; mood stabilizer: milder = 1/30, severe = 13/155, $z = 0.96$, $P > 0.34$). Since we confirmed that there was no significant difference in the classification performance between the 2 groups, these results suggest that our brain network marker is not derived from the effects of different doses of antidepressants on brain circuits. Moreover, all patients with MDD at the KUT imaging site, which is included in the discovery dataset, were treatment resistant (treatment-resistant depression: adequate use of 2 or more antidepressants for 4 to 6 weeks is not efficacious, or intolerance to 2 or more antidepressants exists). We calculated the classification accuracy only at KUT and obtained the same performance level (accuracy = 71%). These results suggest that the current MDD classifier can be generalized to milder depression, as well as to treatment-resistant patients with MDD.

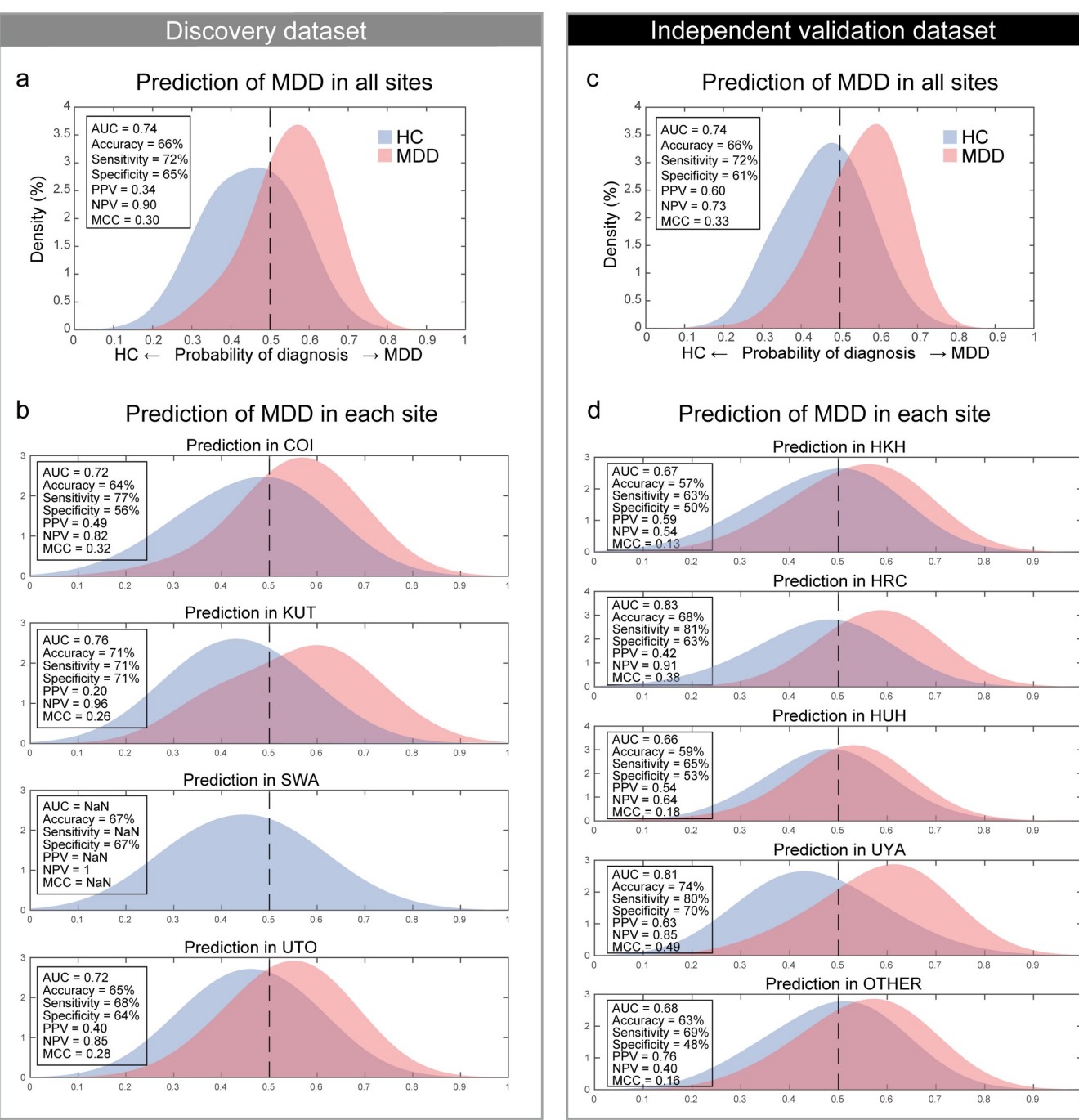

**Fig 3. MDD classifier performances in both datasets. (a)** The probability distribution for the diagnosis of MDD in the discovery dataset and **(b)** probability distributions for each imaging site. MDD and HC distributions are depicted in red and blue, respectively. **(c)** The probability distribution for the diagnosis of MDD in the independent validation dataset and **(d)** probability distributions for each imaging site. MDD and HC distributions are depicted in red and blue, respectively. The numerical data used in this figure are included in S1 Data. AUC, area under the curve; COI, Center of Innovation in Hiroshima University; HC, healthy control; HKH, Hiroshima Kajikawa Hospital; HRC, Hiroshima Rehabilitation Center; HUH, Hiroshima University Hospital; KUT, Kyoto University; MCC, Matthews correlation coefficient; MDD, major depressive disorder; NPV, negative predictive value; PPV, positive predictive value; SWA, Showa University; UTO, University of Tokyo; UYA, Yamaguchi University. OTHER, the data from OpenNeuro (https://openneuro.org/datasets/ds002748/versions/1.0.0).

Furthermore, we investigated the correlation between the classifier's output (probability of depression) and depressive symptoms (BDI score). We found significant correlations in both the discovery dataset and validation dataset (discovery dataset: $r = 0.29$, $P < 1.0 \times 10^{-10}$, validation dataset: $r = 0.32$, $P < 1.0 \times 10^{-11}$). However, we were not able to find consistently significant correlations within each group (HC and MDD groups) in both datasets (discovery dataset: MDD [$r = -0.07$, $P = 0.46$], HC [$r = 0.13$, $P = 0.010$]; validation dataset: MDD [$r = 0.0043$, $P = 0.95$], HC [$r = -0.13$, $P = 0.04$]).

To rule out the possibility that the classifier's performance is driven by age, we constructed our brain network marker using the datasets in which the mean age was matched between MDD and HC groups. We further assessed whether we can predict classifier scores (probability of depression) based on age, sex, and the amount of motion (frame-wise displacement [FD]) or the combination of head movement parameters ($x$, $y$, $z$, yaw, pitch, and roll), respectively [44]. As a result, we confirmed that we could not predict classifier scores based on age, sex, and the amount of motion (FD) or the combination of head movement parameters (S2 Text and S1 Fig). These results indicate that the classifier's performance was unlikely to have been driven by confounds.

Regarding the effectiveness of the developed network marker, although discriminability was acceptable (AUC = 0.74) in the independent validation dataset, the performance of the PPV was low in the discovery dataset (0.34). This occurred because the number of patients with MDD was much smaller than that of HCs (about 4 times as many HCs as MDDs) in the discovery dataset. By contrast, in the independent validation dataset, in which the number of HCs was about 1.5 times as high as the number of MDDs, the PPV, at 0.60, was acceptable. When applying a developed network marker in clinical practice, we assume this marker to be applied to those who actually visit the hospital. Therefore, the actual PPV will be acceptable in clinical practice because the prevalence of MDD may be relatively higher compared with the prevalence of MDD in the general population. Furthermore, in the independent validation dataset, when we divided the dataset into low- and high-risk groups based on the cutoff value (probability of MDD being 0.52) determined in the discovery dataset [45], the odds (sensitivity/1 − sensitivity) were 1.92 in the high-risk group. Moreover, the odds ratio was 4.95 when the odds in the low group were set to 1. That is, the output of the classifier (probability of MDD) will be useful information for psychiatrists as a physical measure supplementing patients' symptoms and signs in order to make a diagnosis. We checked the stability of our developed network marker to assess whether the same subject was consistently classified into the same class when the subject was scanned multiple times at various imaging sites. We applied our marker to a traveling subject dataset in which 9 healthy participants (all male participants; age range, 24 to 32 years; mean age, 27 ± 2.6 years) were scanned about 50 times at 12 different sites, producing a total of 411 scans (S3 Table). We achieved a high accuracy in this dataset (mean accuracy = 84.5, 1SD = 12.8, across participants). This result indicates that our developed network marker has high stability even if the same subject is scanned multiple times at various imaging sites.

To objectively compare and verify the stability of diagnosis by our brain network marker with that of the diagnosis by clinicians based on the DSM, we performed a new simulation. In this simulation, we prepared a surrogate set of 1,000 participants who have a "true" probability of depression ($0 < p(MDD) < 1$). In reality, we used the classifier's outputs for real patients from our datasets as the true probability. We assumed that the rater's noise ($N(0, \Sigma)$) was added to the probability of depression ($\hat{p}(MDD)$), and a participant with $\hat{p}(MDD) = 0.5$ or more was diagnosed with MDD. By assuming 2 raters, we can calculate a diagnostic agreement rate between these 2 raters' diagnoses (kappa). Based on this simulation, we estimated the rater's variance $\Sigma = 0.0149$ when kappa is 0.28 [28,29]. Next, we assessed the variance of our

classifier's output across imaging sites. We applied our classifier to a traveling subject dataset. We evaluated the inter-rater variance of our classifier's output assuming that each MRI scanner was a rater. We estimated whole-variance (variance across all about 50 scans), between-sites-variance (we calculated the average of 2 or 3 scans per imaging site and then took the variance across the average values), and within-site-variance (variance across 15 scans at 1 imaging site) in each participant. As a result, the whole-variance was 0.0061, the between-sites-variance was 0.0037, and the within-site-variance was 0.0055 averaged across participants. These values were about 25%~40% of the variance between clinicians in the above simulation. These results indicate that the diagnosis by our brain network marker could be more objective and stable than the diagnosis by clinicians, even considering the variance across fMRI scanners and imaging sites.

To assess the effects of harmonization, we compared prediction performances and the number of selected FCs among brain network markers constructed with harmonization and without harmonization for the discovery dataset and the independent validation dataset, respectively. As a result, we found significant improvements in the prediction performance by harmonization for the independent validation dataset, but not by harmonization for the discovery dataset (S3 Text, S4 Table, and S2 Fig). One possible explanation for which the harmonization for the discovery dataset did not improve the classification performance was that the site effect may have been sufficiently small in the discovery dataset since the discovery dataset was acquired using a unified imaging protocol. However, this was not the case in the validation dataset. A detailed analysis on how classification performance with/without harmonization depends on the size and patterns of the site and disease difference would make for an interesting future research topic. On the other hand, the number of selected FCs was the largest for traveling subject harmonization (25 FCs), compared with 23 and 21 without harmonization and ComBat, respectively (we explained how to select important FCs for MDD diagnosis in important FCs for MDD diagnosis in the Results section) (S3 Text, S4 Table, and S2 Fig). This suggests that we could extract more brain circuit information with regard to MDD from the data by the traveling subject harmonization.

We investigated whether the discrimination performances were different across imaging sites in the independent validation dataset. We calculated the 95% confidence intervals (CIs) of the discrimination performances (AUC, accuracy, sensitivity, and specificity) using a bootstrap method for every imaging site. We repeated the bootstrap procedure 1,000 times and calculated the 95% CI for each site. We then checked whether there was a site whose CI did not overlap with the CIs of other imaging sites. We were unable to find such an imaging site, suggesting no significant systematic difference (S4 Text and S3 Fig). We further assessed the prediction performance when we used a parcellation scheme other than Glasser's region of interest (ROI). We found that there was no large difference in the prediction performance dependent on ROI numbers or parcellation schemes (S5 Text and S4 Fig).

## Important FCs for MDD diagnosis

We examined important resting-state FCs for an MDD diagnosis. Briefly, we counted the number of times an FC was selected by LASSO during the 10-fold cross-validation (CV). We considered this FC to be important if this number was significantly higher than the threshold for randomness, according to a permutation test. We permuted the diagnostic labels of the discovery dataset and conducted a 10-fold CV and 10-subsampling procedure, and we repeated this permutation procedure 100 times. We then used the number of counts for each connection selected by the sparse algorithm during 10 subsamplings × 10-fold CV (max 100 times) as a statistic in every permutation dataset. To control for the multiple comparison problem, we

set a null distribution as the max distribution of the number of counts over all FCs and set our statistical significance to a certain threshold (permutation test, $P < 0.05$, 1-sided).

Fig 4A shows the spatial distribution of the 25 FCs that were automatically and unbiasedly identified from the data for the reliable classification of MDD and HC by the machine learning algorithms. We hereafter summarize the characteristics of these FCs. (1) The FC between the left and right insula had the largest difference between patients with MDD and HCs (FC#12 in Fig 4B). (2) A total of 19 of 25 FCs exhibited "under-connectivity," and only 6 FCs exhibited "over-connectivity." Note that the state of FC exhibiting the smaller (i.e., more negative) and greater (more positive) mean correlation values in the MDD population than the HC population is termed under- and over-connectivity, respectively. (3) Two FCs (FC#11 and FC#23 in Fig 4B) were related to the subgenual anterior cingulate cortex (sgACC). (4) FC#2 was FC between the sensory motor cortex (postcentral cortex) and the left dorsolateral prefrontal cortex (left DLPFC). A detailed list of the FCs is provided in S5 Table. Furthermore, the mean FC values for HCs and patients with MDD in the discovery dataset (Fig 5A) were similar to those in the independent validation dataset (Fig 5B), pointing to the reproducibility of the FC values.

## Generalization of the MDD classifier to other disorders

We sought to investigate and confirm the spectral structure among the disorders as revealed by previous studies [16–18]. If the MDD classifier predicts patients with a different disorder from patients from MDD, then the probability of diagnosis for patients with that disorder should be over 0.5. In this case, we may say that the patients possess some degree of MDD-ness and that this disorder is related to MDD according to the imaging biological dimension. To assess this possibility, we applied our MDD classifier to patients with SCZ and ASD included in the DecNef Project Brain Data Repository (S6 Table, https://bicr-resource.atr.jp/srpbsopen/).

We found that patients with SCZ have high MDD-ness (accuracy = 76%, $P = 2.0 \times 10^{-12}$, 2-way binomial test) and that patients with ASD did not have high MDD-ness (accuracy = 55%, $P = 0.20$, 2-way binomial test) (S6 Text and S5 Fig). This result suggests that the MDD classifier generalizes to SCZ but not to ASD. We note that our discovery dataset for the construction of the MDD classifier did not include any patients with MDD who were comorbid with SCZ and only 1 patient with MDD who was comorbid with ASD. Therefore, our classifier was not affected by either SCZ or ASD diagnosis. Thus, the above generalization of the MDD classifier may point to a certain neurobiological relevance among diseases.

## Discussion

In this study, we thoroughly considered conditions and resolved difficulties in order to ensure the generalization of our brain network marker in the independent validation dataset, which does not include any imaging sites of the discovery dataset. We succeeded in generalizing our network marker to the big independent validation dataset. This generalization ensures scientific reproducibility and the clinical applicability of rs-fMRI. Without this fundamental evidence, we cannot proceed to the development of rs-fMRI-based subtyping, evaluation of drug effects, or exploration of a multi-spectrum disorder in the biological dimensions, as mentioned in the Introduction section. Therefore, our study found generalizable psychiatric biomarkers, which the fields of psychiatry, neuroscience, and computational theory have long sought out, to no avail, since the RDoC initiative.

We developed generalizable brain network markers without restriction to treatment-resistant or melancholic MDD. Most previous studies have reported the performance of a prediction model using data from the same imaging sites using a CV technique. However, because of

### a  Important functional connections for MDD diagnosis

FPN    DMN    Motor    Salience    Auditory    Uncertain    Visual    Attention    Subcortical

Left hemisphere          Back view          Right hemisphere          Top view

### b

**Fig 4. Important FCs for MDD diagnosis. (a)** The 25 FCs viewed from left, back, right, and top. Interhemispheric connections are shown in the back and top views only. Regions are color-coded according to the intrinsic network. The state of functional connectivity exhibiting the smaller (i.e., more negative) and greater (more positive) mean correlation index in the MDD population than in the HC population is termed under- (blue line) and over-connectivity (red line), respectively. The width of the line represents the effect size of the difference (*t*-value) in the FC values between MDD and HC groups. **(b)** Listed here are the laterality and anatomical identification of the ROI as identified by the AAL and associated intrinsic networks related to the 25 FCs. AAL, anatomical automatic labeling; DMN, default mode network; FC, functional connectivity; FPN, fronto-parietal network; HC, healthy control; MDD, major depressive disorder; ROI, region of interest. An interactive display of each connection in Fig 4A can be accessed from https://bicr.atr.jp/~imamizu/p5/MDD/.

large imaging site differences in rs-fMRI data [26,46], CV methods generally induce inflations in performance. To ensure reproducibility, it is critical to demonstrate the generalizability of the models with an independent validation dataset acquired from completely different imaging sites [15,20–22]. To overcome the abovementioned site differences, we reduced site differences in a multisite large-scale rs-fMRI dataset using our novel harmonization method. Next, we constructed an MDD classifier that was acceptably generalized to the independent validation dataset. Acceptable generalized prediction performance was also achieved for the 5 individual imaging site datasets (Fig 3D). This generalization was achieved even though the imaging protocols in the independent validation datasets were different from the discovery dataset. There

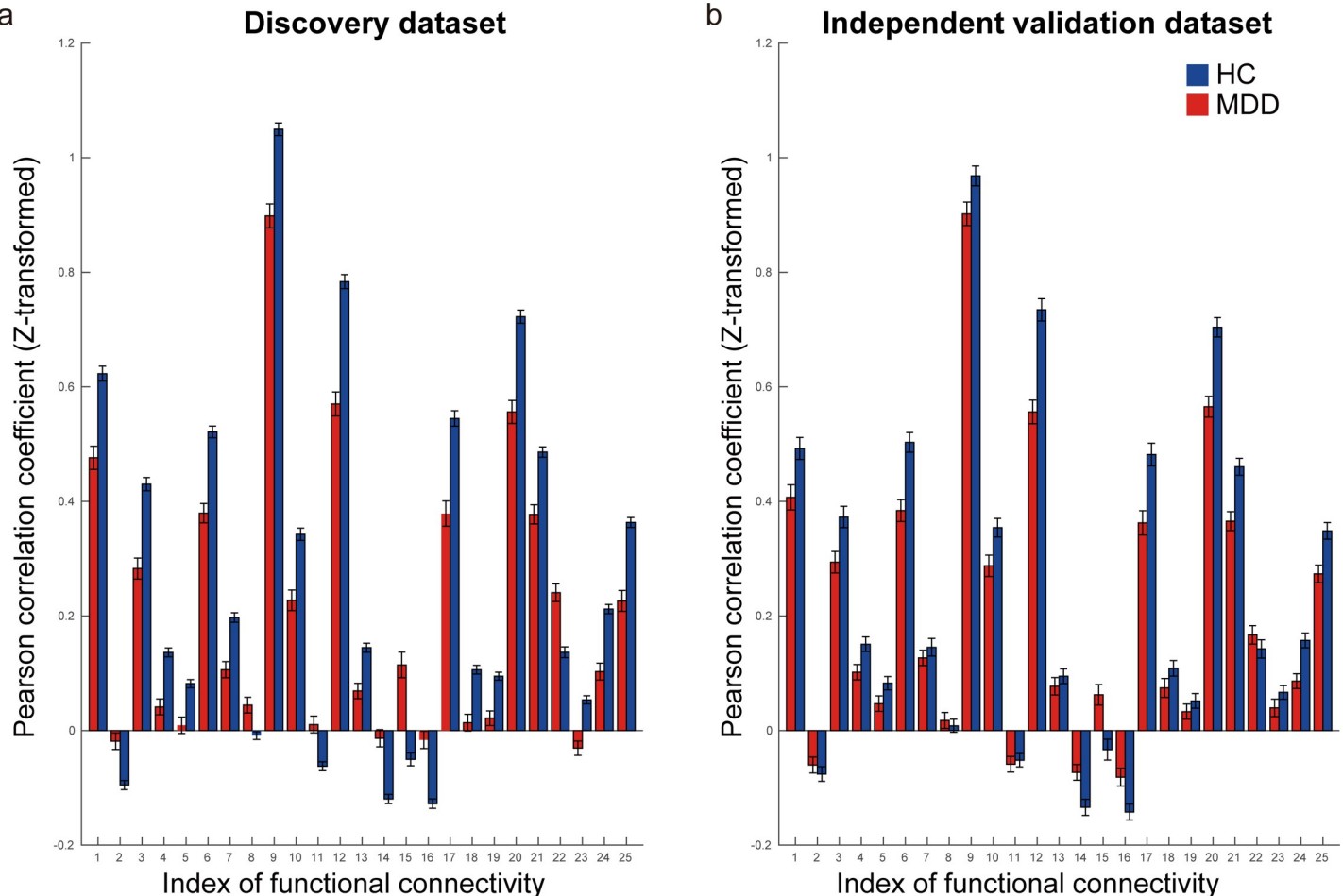

**Fig 5. Reproducibility of important FCs for MDD diagnosis. (a)** The FC values for both HCs (blue bar) and patients with MDD (red bar) in the discovery dataset. **(b)** The FC values for both HCs and patients with MDD in the independent validation dataset. The error bar represents the standard error. The numerical data used in this figure are included in S1 Data. FC, functional connectivity; HCs, healthy controls; MDD, major depressive disorder.

are only 2 studies in which generalization of FC-based MDD classifiers to independent validation data was demonstrated [12,16]. To the best of our knowledge, our work is the first to construct a generalized classifier of MDD without restriction to certain MDD subtypes: Drysdale concentrated on patients with MDD who were treatment resistant, and Ichikawa restricted patients with the melancholic subtype of MDD. Constructing the whole MDD marker is important for subsequent MDD subtyping analyses. This was achieved for the first time by collecting data on a large variety of patients with MDD from multiple imaging sites and objectively harmonizing them with a traveling subject dataset. Furthermore, our simulation result indicates that our brain network marker of MDD could more objectively and stably diagnose MDD than the diagnosis by clinicians based on the DSM, even in consideration of the variance across fMRI scanners and imaging sites.

For the application of our harmonization method to the actual medical field, the best classification performance will be achieved when (1) all imaging sites are involved in a traveling subject dataset; (2) the traveling subject harmonization is applied throughout the discovery and the validation datasets; and (3) a brain network marker is reconstructed from the harmonized datasets. However, when we use our method under a medical device program approval, it seems more realistic to conduct the harmonization without using the discovery dataset because imaging sites cannot be added to the dataset after the approval. As such, the method in this paper or the method of applying the traveling subject harmonization within the independent validation dataset should be preferred. It is important to note that because of the absence of the traveling subject dataset, traveling subject harmonization was not possible for the independent validation dataset, and we were forced to use ComBat in this case.

The machine learning algorithms reliably identified the 25 FCs that are important for MDD diagnosis (Fig 4 and S5 Table). We hereafter summarize the characteristics of these FCs. (1) The FC between the left and right insula revealed the largest differences between patients with MDD and HCs (FC#12 in Fig 4B). Abnormalities in the insula were not only found in patients with MDD [47,48] but also reported as common abnormalities (reduced gray matter volume) among psychiatric disorders [4]. Therefore, the connectivity associated with the insula is a potential candidate for the neurobiological dimension to understand a multi-spectrum disorder. (2) A total of 19 of 25 FCs exhibited "under-connectivity," and only 6 FCs exhibited "over-connectivity." Note that the state of FC exhibiting the smaller (i.e., more negative) and greater (more positive) mean correlation values in the MDD population than the HC population is termed under- and over-connectivity, respectively. (3) Two FCs (FC#11 and FC#23 in Fig 4B) were related to the sgACC. According to a previous study, sgACC is metabolically overactive in treatment-resistant depression and is known as an important treatment target of deep brain stimulation for MDD [49]. (4) FC#2 was the FC between the sensory motor cortex (postcentral cortex) and the left DLPFC. Previous study shows that the left DLPFC is anticorrelated with the sgACC and is known as an important treatment target of repetitive transcranial magnetic stimulation (rTMS) for MDD [50]. These results indicate that we need further analyses to clarify how the classifier's output and abnormalities in each FC are associated with cognitive and affective functions in a future study.

Ultimately, it would be very important to understand the relationships across disorders (multi-disorder spectrum). We found that SCZ had a high tendency (similarity) toward MDD, while ASD had no such a tendency toward MDD (S5 Fig). This result suggests that the MDD classifier generalizes to SCZ but not to ASD. Thus, the above generalization of the MDD classifier may point to a certain neurobiological relevance among diseases. Our patients with SCZ were in the chronic phase and had negative symptoms. Considering that the negative symptoms of SCZ are similar to those of depression [51–56], the generalization hypothesizes the existence of neurobiological dimensions underlying the common symptoms between SCZ and

MDD. We need further analyses to quantitatively examine the neurobiological relationship between SCZ and MDD by gathering more precise information on SCZ (symptoms and medication history). To further understand the multi-disorder spectrum, we developed markers of SCZ and ASD using the same method as in this study in addition to a brain network marker of MDD (S6 Text). As a result, we found an interesting asymmetric relationship among these disorders: The classifier of SCZ did not generalize to patients with MDD (S5 Fig). This kind of asymmetry in the classifiers had also been found between the SCZ classifier and the ASD classifier (the ASD classifier generalized to SCZ, but the SCZ classifier did not generalize to ASD) [17,18]. These results provide us with important information for understanding the biological relationships between diseases. For example, the above asymmetry between the SCZ and ASD or MDD classifiers suggests that the brain network related to SCZ is characterized by a larger diversity than that of ASD or MDD and that it partially shares information with the smaller brain network related to ASD or MDD than that of SCZ [18].

Although biomarkers have been developed with the aim of diagnosing patients, the focus has shifted to the identification of biomarkers that can determine therapeutic targets, such as theranostic biomarkers [57,58], which would allow for more personalized treatment approaches. The 25 FCs discovered in this study are promising candidates as theranostic biomarkers for MDD because they are related to the MDD diagnosis. Future work should investigate whether modulation of FC could be an effective treatment in MDD by using an intervention method with regard to FC, such as functional connectivity neurofeedback training [57–61].

## Materials and methods

### Ethics statement

All participants in all datasets provided written informed consent. All recruitment procedures and experimental protocols were approved by the institutional review boards of the principal investigators' respective institutions (Advanced Telecommunications Research Institute International [approval numbers: 13–133, 14–133, 15–133, 16–133, 17–133, and 18–133], Hiroshima University [E-38], Kyoto Prefectural University of Medicine [RBMR-C-1098], Showa University [SWA] [B-2014-019 and UMIN000016134], the University of Tokyo [UTO] Faculty of Medicine [3150], Kyoto University [C809 and R0027], and Yamaguchi University [H23-153 and H25-85]) and conducted in accordance with the Declaration of Helsinki.

### Participants

We used 2 rs-fMRI datasets for the analyses: (1) The "discovery dataset" contained data from 713 participants (564 HCs from 4 sites, 149 patients with MDD from 3 sites; Table 1). Each participant underwent a single rs-fMRI session, which lasted for 10 min. Within the Japanese SRPBS DecNef project, we planned to acquire the rs-fMRI data using a unified imaging protocol (S1 Table; http://bicr.atr.jp/rs-fmri-protocol-2/). However, there were 2 erroneous phase-encoding directions (P→A and A→P). In addition, different sites had different MRI hardware (S1 Table). During the rs-fMRI scans, participants were instructed to "Relax. Stay Awake. Fixate on the central crosshair mark, and do not concentrate on specific things." This dataset was acquired in the SRPBS DecNef project from 2014. (2) The "independent validation dataset" contained data from 449 participants (264 HCs and 185 patients with MDD from 4 independent sites; Table 1). Data were acquired following protocols reported in S1 Table. The sites used were different from the discovery dataset. Each participant underwent a single rs-fMRI session lasting for 5 or 8 min. This data set was acquired in other projects from 2008 and not the SRPBS DecNef. The dataset collected from Hiroshima University Hospital (HUH),

Hiroshima Kajikawa Hospital (HKH), and Hiroshima Rehabilitation Center (HRC) in the independent validation dataset were acquired by "development of diagnosis and treatment techniques for patients with severe intractable depression and insensitivity to antidepressant treatment based on molecular and cellular researches on BDNF and depression" of the Japan Science and Technology Agency Core Research for Evolutional Science and Technology (CREST) from 2008 and by "understanding the neurocircuit–molecular mechanism underlying pathophysiology of depression and the development of its neuroscience-based diagnosis and treatment" of the SRPBS from 2011. The dataset collected from Yamaguchi University (UYA) was acquired by "exploration of the biological markers for discrimination of heterogeneous pathophysiology of major depressive disorder" of the SRPBS from 2012. We further included the dataset collected from a country outside Japan (OpenNeuro: https://openneuro.org/datasets/ds002748/versions/1.0.0) in the final independent validation dataset (21 HCs and 51 patients with MDD; Table 1). In both datasets, depression symptoms were evaluated using the BDI-II score obtained from most participants in each dataset. This study was carried out in accordance with the recommendations of the institutional review boards of the principal investigators' respective institutions (Hiroshima University, Kyoto University, Showa University, University of Tokyo, and Yamaguchi University) with written informed consent from all subjects in accordance with the Declaration of Helsinki. The protocol was approved by the institutional review boards of the principal investigators' respective institutions (Hiroshima University, Kyoto University, Showa University, University of Tokyo, and Yamaguchi University). Most data utilized in this study can be downloaded publicly from the DecNef Project Brain Data Repository at https://bicr-resource.atr.jp/srpbsopen/, https://bicr.atr.jp/dcn/en/download/harmonization/, and https://openneuro.org/datasets/ds002748/versions/1.0.0. The data availability statements of each site are described in S1 Table.

## Preprocessing and calculation of the resting-state FC matrix

We preprocessed the rs-fMRI data using FMRIPREP version 1.0.8 [62]. The first 10 s of the data were discarded to allow for T1 equilibration. Preprocessing steps included slice-timing correction, realignment, coregistration, distortion correction using a field map, segmentation of T1-weighted structural images, normalization to Montreal Neurological Institute (MNI) space, and spatial smoothing with an isotropic Gaussian kernel of 6 mm full width at half maximum. "Fieldmap-less" distortion correction was performed for the independent validation dataset due to the lack of field map data. For more details on the pipeline, see http://fmriprep.readthedocs.io/en/latest/workflows.html. For 6 participants' data in the independent validation dataset, the coregistration was unsuccessful, and we therefore excluded these data from further analysis.

## Parcellation of brain regions

To analyze the data using Human Connectome Project (HCP) style surface-based methods, we used ciftify toolbox version 2.0.2 [63]. This allowed us to analyze our data, which lacked the T2-weighted image required for HCP pipelines, using an HCP-like surface-based pipeline. Next, we used Glasser's 379 surface-based parcellations (cortical 360 parcellations + subcortical 19 parcellations) as ROIs, considered reliable brain parcellations [34]. The BOLD signal time courses were extracted from these 379 ROIs. To facilitate the comparison of our results with previous studies, we identified the anatomical names of important ROIs and the names of intrinsic brain networks that included the ROIs using anatomical automatic labeling (AAL) [64] and Neurosynth (http://neurosynth.org/locations/).

## Physiological noise regression

Physiological noise regressors were extracted by applying CompCor [65]. Principal components were estimated for the anatomical CompCor (aCompCor). A mask to exclude signals with a cortical origin was obtained by eroding the brain mask and ensuring that it contained subcortical structures only. Five aCompCor components were calculated within the intersection of the subcortical mask and the union of the cerebrospinal fluid (CSF) and white matter (WM) masks calculated in the T1-weighted image space after their projection to the native space of functional images in each session. To remove several sources of spurious variance, we used a linear regression with 12 regression parameters, such as 6 motion parameters, average signals over the whole brain, and 5 aCompCor components.

## Temporal filtering

A temporal bandpass filter was applied to the time series using a first-order Butterworth filter with a pass band between 0.01 Hz and 0.08 Hz to restrict the analysis to low-frequency fluctuations, which are characteristic of rs-fMRI BOLD activity [66].

## Head motion

FD [67] was calculated for each functional session using Nipype (https://nipype.readthedocs.io/en/latest/). FD was used in the subsequent scrubbing procedure. To reduce spurious changes in FC from head motion, we removed volumes with FD >0.5 mm, as proposed in a previous study [67]. The FD represents head motion between 2 consecutive volumes as a scalar quantity (i.e., the summation of absolute displacements in translation and rotation). Using the aforementioned threshold, 6.3% ± 13.5 volumes (mean ± SD) were removed per rs-fMRI session in all datasets. If the ratio of the excluded volumes after scrubbing exceeded the mean + 3 SD, participants were excluded from the analysis. As a result, 32 participants were removed from all datasets. Thus, we included 683 participants (545 HCs, 138 patients with MDD) in the discovery dataset and 440 participants (259 HCs, 181 patients with MDD) in the independent validation dataset for further analysis.

## Calculation of FC matrix

FC was calculated as the temporal correlation of rs-fMRI BOLD signals across 379 ROIs for each participant. There are a number of different candidates to measure FC, such as the tangent method and partial correlation; however, we used a Pearson's correlation coefficient because they are the most commonly used values in previous studies. Fisher's z-transformed Pearson's correlation coefficients were calculated between the preprocessed BOLD signal time courses of each possible pair of ROIs and used to construct $379 \times 379$ symmetrical connectivity matrices in which each element represents a connection strength between 2 ROIs. We used 71,631 FC values [$(379 \times 378)/2$] of the lower triangular matrix of the connectivity matrix for further analysis.

## Control of site differences

Next, we used a traveling subject harmonization method to control for site differences in FC in the discovery dataset. This method enabled us to subtract pure site differences (measurement bias), which are estimated from the traveling subject dataset wherein multiple participants travel to multiple sites to assess measurement bias. The participant factor ($p$), measurement bias ($m$), sampling biases ($s_{hc}$, $s_{mdd}$), and psychiatric disorder factor ($d$) were estimated by fitting the regression model to the FC values of all participants from the discovery dataset and

the traveling subject dataset. For each connectivity, the regression model can be written as fol-
lows:

$$Connectivity = \mathbf{x}_m^\mathrm{T} \boldsymbol{m} + \mathbf{x}_{s_{hc}}^\mathrm{T} \boldsymbol{s}_{hc} + \mathbf{x}_{s_{mdd}}^\mathrm{T} \boldsymbol{s}_{mdd} + \mathbf{x}_d^\mathrm{T} \boldsymbol{d} + \mathbf{x}_p^\mathrm{T} \boldsymbol{p} + const + e,$$

such that $\sum_j^9 p_j = 0, \sum_k^4 m_k = 0, \sum_k^4 s_{hck} = 0, \sum_k^3 s_{mddk} = 0, d_1(\mathrm{HC}) = 0,$

in which $\boldsymbol{m}$ represents the measurement bias (4 sites × 1), $\boldsymbol{s}_{hc}$ represents the sampling bias of
HCs (4 sites × 1), $\boldsymbol{s}_{mdd}$ represents the sampling bias of patients with MDD (3 sites × 1), $\boldsymbol{d}$ repre-
sents the disorder factor (2 × 1), $\boldsymbol{p}$ represents the participant factor (9 traveling subjects × 1),
$const$ represents the average FC value across all participants from all sites, and $e \sim \mathcal{N}(0, \gamma^{-1})$
represents noise. Measurement biases were removed by subtracting the estimated measure-
ment biases. Thus, the harmonized FC values were set as follows:

$$Connectivity^{Harmonized} = Connectivity - \mathbf{x}_m^\mathrm{T} \hat{\boldsymbol{m}},$$

in which $\hat{\boldsymbol{m}}$ represents the estimated measurement bias. More detailed information have been
previously described [26].

   We used the ComBat harmonization method [35–38] to control for site differences in FC
in the independent validation dataset because we did not have a traveling subject dataset for
those sites. We performed harmonization to correct only for the site difference using informa-
tion on MDD diagnosis, BDI score, age, sex, and dominant hand as auxiliary variables in Com-
Bat. Notably, compared with the conventional regression method, the ComBat method is a
more advanced method to control for site effects [35–38].

## Constructing the MDD classifier using the discovery dataset

We constructed a brain network marker for MDD that distinguished between HCs and
patients with MDD using the discovery dataset based on 71,631 FC values. To construct the
network marker, we applied a machine learning technique. Although SVM is often used as a
classifier, SVM is not suitable for investigating the contribution of explanatory variables
because it is difficult to calculate the contribution of each explanatory variable. Based on our
previous study [17], we assumed that psychiatric disorder factors were not associated with
whole brain connectivity, but rather with a specific subset of connections. Therefore, we con-
ducted logistic regression analyses using the LASSO method to select the optimal subset of FCs
[40]. A logistic function was used to define the probability of a participant belonging to the
MDD class as follows:

$$P_{sub}(y_{sub} = 1 | \boldsymbol{c}_{sub}; \boldsymbol{w}) = \frac{1}{1 + \exp(-\boldsymbol{w}^\mathrm{T} \boldsymbol{c}_{sub})},$$

in which $\boldsymbol{y}_{sub}$ $y_{sub}$ represents the class label (MDD, $y = 1$; HC, $y = 0$) of a participant, $\boldsymbol{c}_{sub} c_{sub}$
represents an FC vector for a given participant, and $w$ represents the weight vector. The weight
vector $w$ was determined to minimize

$$J(\mathbf{w}) = -\frac{1}{n_{sub}} \sum_{j=1}^{n_{sub}} \log P_j(y_j = 1 | \boldsymbol{c}_j; \boldsymbol{w}) + \lambda \|\boldsymbol{w}\|_1,$$

in which $\|\boldsymbol{w}\|_1 = \sum_i^N |w_i|$ and $\lambda$ represent hyperparameters that control the amount of shrink-
age applied to the estimates. To estimate weights of the logistic regression and a hyperpara-
meter $\lambda$, we conducted a nested cross-validation procedure (Fig 2). In this procedure, we first

divided the whole discovery dataset into a training set (9 folds of 10 folds), which used for training a model and a test set (a fold of 10 folds) for testing the model. To minimize bias due to the differences in the numbers of patients with MDD and HCs, we used an undersampling method [41]. Almost 125 patients with MDD and 125 HCs were randomly sampled from the training set, and the classifier performance was tested using the test set. When we performed the undersampling and subsampling procedures, we matched the mean age between MDD and HC groups in each subsample. Since only a subset of training data is used after undersampling, we repeated the random sampling procedure 10 times (i.e., subsampling). We then fitted a model to each subsample while tuning a regularization parameter in the inner loop of the nested cross-validation, resulting in 10 classifiers. For the inner loop, we used the "*lassoglm*" function in MATLAB (R2016b, Mathworks, USA) and set "NumLambda" to 25 and "CV" to 10. In this inner loop, we first calculated a value of λ just large enough such that the only optimal solution is the all-zeroes vector. A total of 25 values of λ were prepared at equal intervals from 0 to $\lambda_{max}$, and the λ was determined according to the one standard error rule in which we selected the largest λ within the standard deviation of the minimum prediction error (among all λ) [27]. The mean classifier output value (diagnostic probability) was considered indicative of the classifier output. Diagnostic probability values of >0.5 were considered indicative of patients with MDD. We calculated the AUC using the "*perfcurve*" function in MATLAB. In addition, we calculated the accuracy, sensitivity, specificity, PPV, and NPV. Furthermore, we evaluated classifier performance for the unbalanced dataset using the MCC [42,43], which takes into account the ratio of the confusion matrix size.

## Generalization performance of the classifier

We tested the generalizability of the network marker using an independent validation dataset. We created 100 classifiers of MDD (10-fold CV × 10 subsamples); therefore, we applied all trained classifiers to the independent validation dataset. Next, we averaged the 100 outputs (diagnostic probability) for each participant and considered the participant to be a patient with MDD if the averaged diagnostic probability value was >0.5.

To test the statistical significance of the MDD classifier performance, we performed a permutation test. We permuted the diagnostic labels of the discovery dataset and conducted a 10-fold CV and 10-subsampling procedure. Next, we took an average of the 100 outputs (diagnostic probability); a mean diagnostic probability value of >0.5 was considered indicative of a diagnosis of MDD. We repeated this permutation procedure 100 times and calculated the AUC and MCC as the performance metrics of each permutation.

## Identification of the important FCs linked to diagnosis

We examined important resting-state FC for an MDD diagnosis. Briefly, we counted the number of times an FC was selected by LASSO during the 10-fold CV. We considered that this FC was important if this number was significantly higher than chance, according to a permutation test. We permuted the diagnostic labels of the discovery dataset and conducted a 10-fold CV and 10-subsampling procedure and repeated this permutation procedure 100 times. We then used the number of counts for each connection selected by the sparse algorithm during 10-fold CVs × 10 subsamplings (max 100 times) as a statistic in every permutation dataset. To control for the multiple comparison problem, we set a null distribution as the max distribution of the number of counts over all functional connections and set our statistical significance to a certain threshold ($P < 0.05$, 1-sided). FCs selected ≥17 times out of a total of 100 times were regarded as diagnostically important.

## Supporting information

**S1 Data. Excel spreadsheet containing, in separate sheets, the underlying numerical data for Figs 1, 3A–3D, 5A and 5B, and all Supporting information figures.**
(XLSX)

**S1 Text. Prediction performance using SVM.**
(DOCX)

**S2 Text. Analysis and validation of controls for confound artifact.**
(DOCX)

**S3 Text. Utility of harmonization.**
(DOCX)

**S4 Text. Differences in prediction performance among imaging sites.**
(DOCX)

**S5 Text. Higher/lower resolution of regions of interest.**
(DOCX)

**S6 Text. Generalization of the classifiers to other disorders.**
(DOCX)

**S1 Fig. Distribution of prediction performance and difference in mean age between MDD and HC across all subsamples.** The distribution of the AUC and *t*-value (difference in mean age between MDD and HC groups) across all subsamples. The numerical data used in this figure are included in S1 Data. AUC, area under the curve; HC, healthy control; MDD, major depressive disorder.
(TIF)

**S2 Fig. Comparing the prediction performances among harmonization schemes.** (**a**) The prediction performance (AUC) of the MDD classifier in the independent validation dataset for each harmonization scheme for the discovery dataset and the independent validation dataset (without harmonization; blue bar, ComBat harmonization; yellow bar). (**b**) Probability distributions for the diagnosis of MDD in the data from OpenNeuro (OTHER) without harmonization or with ComBat harmonization. The numerical data used in this figure are included in S1 Data. AUC, area under the curve; HC, healthy control; MCC, Matthews correlation coefficient; MDD, major depressive disorder.
(TIF)

**S3 Fig. Bootstrap prediction performances in the independent validation dataset.** Prediction performances of the MDD classifier in the independent validation dataset in each site. Each color bar indicates a site. Error bar shows the 95% confidence interval from the bootstrap. The numerical data used in this figure are included in S1 Data. AUC, area under the curve; HKH, Hiroshima Kajikawa Hospital; HRC, Hiroshima Rehabilitation Center; HUH, Hiroshima University Hospital; MDD, major depressive disorder; UYA, Yamaguchi University.
(TIF)

**S4 Fig. Higher/lower resolution of regions of interest.** The prediction performances (AUC, accuracy, specificity, and sensitivity) of the MDD classifier constructed by Schaefer's ROIs as a function of the number of ROIs. The numerical data used in this figure are included in S1 Data. AUC, area under the curve; MDD, major depressive disorder; ROI, region of interest.
(TIF)

**S5 Fig. Generalization of the classifiers to other psychiatric disorders.** Density distributions of the probability of diagnosis obtained by applying (a) the MDD marker, (b) SCZ marker, and (c) ASD marker to the HCs and patients with MDD, SCZ, and ASD. In each panel, the patient distribution and the HC distribution are plotted separately, with the colored areas representing one or the other. The numbers in parentheses next to HC, MDD, ASD, and SCZ in each panel indicate the number of subjects in the distributions. The independent validation dataset was used in a and b. HCs in a, b, and c were scanned at the same sites as their corresponding patient data. The numerical data used in this figure are included in S1 Data. ASD, autism spectrum disorder; HC, healthy control; MDD, major depressive disorder; SCZ, schizophrenia.
(TIF)

**S1 Table. Imaging protocols for resting-state fMRI in both datasets.**
(XLSX)

**S2 Table. Clinical characteristics of major depressive disorder patients in the discovery dataset.**
(XLSX)

**S3 Table. Imaging protocols for resting-state fMRI in the traveling subject dataset.**
(XLSX)

**S4 Table. Prediction performances in the independent validation dataset for different harmonization schemes.**
(XLSX)

**S5 Table. Description of important FCs.**
(XLSX)

**S6 Table. Demographic characteristics of participants in both datasets.**
(XLSX)

## Author Contributions

**Conceptualization:** Ayumu Yamashita, Mitsuo Kawato, Hiroshi Imamizu.

**Data curation:** Takashi Yamada, Noriaki Yahata, Akira Kunimatsu, Naohiro Okada, Takashi Itahashi, Ryuichiro Hashimoto, Hiroto Mizuta, Naho Ichikawa, Masahiro Takamura, Go Okada, Hirotaka Yamagata, Kenichiro Harada, Koji Matsuo, Saori C. Tanaka, Kiyoto Kasai, Nobumasa Kato, Hidehiko Takahashi, Yasumasa Okamoto.

**Formal analysis:** Ayumu Yamashita.

**Funding acquisition:** Mitsuo Kawato, Hiroshi Imamizu.

**Investigation:** Ayumu Yamashita.

**Methodology:** Ayumu Yamashita, Yuki Sakai, Okito Yamashita.

**Project administration:** Mitsuo Kawato.

**Software:** Ayumu Yamashita.

**Supervision:** Mitsuo Kawato, Hiroshi Imamizu.

**Visualization:** Ayumu Yamashita.

**Writing – original draft:** Ayumu Yamashita, Mitsuo Kawato, Okito Yamashita, Hiroshi Imamizu.

**Writing – review & editing:** Yuki Sakai, Takashi Yamada, Noriaki Yahata, Akira Kunimatsu, Naohiro Okada, Takashi Itahashi, Ryuichiro Hashimoto, Hiroto Mizuta, Naho Ichikawa, Masahiro Takamura, Go Okada, Hirotaka Yamagata, Kenichiro Harada, Koji Matsuo, Saori C. Tanaka, Mitsuo Kawato, Kiyoto Kasai, Nobumasa Kato, Hidehiko Takahashi, Yasumasa Okamoto, Okito Yamashita, Hiroshi Imamizu.

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
