## [Editor Report · Decision Letter 0]

22 Apr 2020

Dear Dr Yamashita, 

Thank you for submitting your manuscript entitled "Generalizable brain network markers of major depressive disorder across multiple imaging sites" for consideration as a Research Article by PLOS Biology.

Your manuscript has now been evaluated by the PLOS Biology editorial staff, as well as by an Academic Editor with relevant expertise, and I am writing to let you know that we would like to send your submission out for external peer review.

Please re-submit your manuscript within two working days, i.e. by Apr 24 2020 11:59PM.

Kind regards,

Gabriel Gasque, Ph.D.,

Senior Editor

PLOS Biology

---

## [Decision Letter · Decision Letter 1]

4 Jun 2020

Dear Dr Yamashita,

Thank you very much for submitting your manuscript "Generalizable brain network markers of major depressive disorder across multiple imaging sites" for consideration as a Research Article at PLOS Biology. Your manuscript has been evaluated by the PLOS Biology editors, by an Academic Editor with relevant expertise, and by three independent reviewers. You will note that reviewer 3, A. Vania Apkarian, has signed his comments. 

In light of the reviews (below), we will not be able to accept the current version of the manuscript, but we would welcome re-submission of a much-revised version that takes into account the reviewers' comments. We cannot make any decision about publication until we have seen the revised manuscript and your response to the reviewers' comments. Your revised manuscript is also likely to be sent for further evaluation by the reviewers.

We expect to receive your revised manuscript within 2 months. 

**IMPORTANT - SUBMITTING YOUR REVISION**

Your revisions should address the specific points made by each reviewer. As you will see, while there is overall enthusiasm by the reviewers, reviewers 1 and 2 agree, although with slightly different tones, that lumping depression as a single diagnostic entity is not correct/useful and that you should try to identify different subgroups/MDD dimensions. In addition, reviewer 3 thinks that several additional analyses are needed to support your claims. Having discussed the comments with the Academic Editor, we do not think you should change the whole approach - but exploring subgroups and correlations with BDI are useful suggestions. We also think that it might be helpful to test some of the Drysdale et al. cohorts (reviewer 3). However, we also appreciate a limitation in the sense that the datasets across the two papers are no longer independent. We would leave it to you to decide how to proceed here.

Independently, the Academic Editor points to two methodological concerns which should be addressed: 

1) how can harmonization be applied to independent test cohorts without information leakage? What would be the procedure for applying this data prospectively to new cohorts? 

2) can you rule out the possibility that the classifier's performance is driven by confounds, particularly head movement? One approach is to test whether classifier scores (i.e., probability of depression) can be predicted from combinations of head movement parameters (e.g., as in Kohoutova et al. 2020, Nat. Protocols).

Please submit the following files along with your revised manuscript:

*Re-submission Checklist*

*Published Peer Review*

*PLOS Data Policy*

*Blot and Gel Data Policy*

Sincerely,

Gabriel Gasque, Ph.D., 

Senior Editor

PLOS Biology

REVIEWS:

Academic Editor:

1) how can harmonization be applied to independent test cohorts without information leakage? What would be the procedure for applying this data prospectively to new cohorts? 

2) can you rule out the possibility that the classifier's performance is driven by confounds, particularly head movement? One approach is to test whether classifier scores (i.e., probability of depression) can be predicted from combinations of head movement parameters (e.g., as in Kohoutova et al. 2020, Nat. Protocols).

Reviewer #1: This interesting study aimed to examine the utility of a method designed to remove site differences in resting state functional connectivity datasets when aiming to identify a classifier distinguishing patients with major depressive disorder (MDD) from healthy controls. The authors report a 69% accuracy in distinguishing the two groups the classifier in an independent dataset. The authors used another dataset of healthy, schizophrenic and ASD patients to determine the specificity of the classifier, and showed that the classifier generalized to schizophrenia but not ASD.

The major strength of the paper is the large number of participants in the discovery dataset and the use of a large, independent dataset for classifier testing. Additional strengths were the that both discovery and testing cohorts comprised data from multiple sites, and the use of a "traveling participant" subgroup to allow assessment of measurement bias across the different sites in the discovery dataset.

The methodology appears to be sound, using established techniques, as well as a novel harmonization method.

I have some points requiring clarification.

A main point is that it is unclear as to how significant the 69% classification accuracy actually is for differentiating between MDD patients and healthy controls in clinical practice, given the relative ease at distinguishing patients with MDD from healthy controls clinically. A potentially more useful classification would be to identify different subgroups/ MDDD dimensions.

It is not clear how systematically patient medication status was considered in sensitivity analyses examining the generalizability of the classifier to milder versus more severe depression (p.9).

Was the undersampling procedure in the discovery dataset analyses adequate to control for the fact that the number of patients with MDD was 4 x smaller than that of healthy controls?

There is no description of the nature of the main FC profiles important for classifying MDD in the Results. Similarly, it is only In the Discussion that the authors note findings regarding schizophrenia and ASD patient analyses. It would be helpful to have brief summary of these important findings in the Results section.

Reviewer #2: I previously reviewed this paper for the other journal (Nat Comms). I found the authors have made additional changes since then. The biggest change I noticed is that they removed the BDI regression model. Also they changed their emphasis from "common brain networks between the classification and regression model" to "generalizable brain markers of MDD". These changes resolved many issues I previously raised, which were related to information leakage and the stability of the feature selection. Given that these problems were among my major concerns in previous reviews, the current version of manuscript seems better than the previous one, but I still find a couple of issues in the current manuscript, though this study certainly has some strengths, including its large sample size collected from multiple sites (N = 1,162; 334 MDD patients) and having a validation set to obtain unbiased estimates of the model performance and its generalizability. 

First, though they started the paper with explaining the RDoC initiative, they still treat the depression as a single diagnostic entity. This lumping approach has been criticized since early 2000 (Hyman, 2008, Nature; Miller, 2010, Science) and the criticism regarding the lumping approach provided the biggest motivation for the RDoC initiative. Depression is a highly heterogeneous condition (Insel & Cuthbert, 2015, Science), and therefore with this type of simple MDD vs. HC classifier, it is difficult to know what the model really captures because the MDD as a whole can mean many different things. They suggested that their model captures the MDD-ness and also anhedonia that is common across MDD and SCZ, but it struck me that the generalization to SCZ not to ASD is not enough to justify this "anhedonia" interpretation because SCZ and MDD shares many features other than anhedonia (e.g., Borsboom & Cramer, 2013). 

Second, in the previous revision, the classification performance for ASD was also significant (p = 2.0 * 10^-7; this is the quote from the previous manuscript, "The separation between ASD individuals and their healthy controls was poorer than that of SCZ but statistically significant (Fig. 8c; AUC = 0.57). "), and I wonder what has been changed since then. One change I noticed that in the previous paper, they reported the AUC, but in the current manuscript, they reported only the accuracy. Considering that this change (sig -> non-sig for ASD classification) seems important for the functional interpretation of the model, the authors should clarify what has been changed and whether the changes can be justified. 

Reviewer #3, A. Vania Apkarian: This is a well executed study hoping to identify resting state fMRI signal that can have utility in identifying subjects with depression. The authors use a large data set from multiple centers, and perform well-organized discovery and validation approach, where the validation data come from separate sources than those used in the discovery. In general the results seem convincing and the author put effort in quantifying the extent to which observed result can be translated to the clinic. There are also important issues that need to be addressed:

1) The paper extensively discusses the utility harmonizing rating state data across different centers. The effect size of this procedure remains unclear and needs to be demonstrated. There is a second worry along this line. The harmonization correction used for discovery and validation data were not the same. How this difference effects outcomes need also further clarification. 

2) There is no evidence that the authors corrected for age and gender influences on FC values. Both of these are well known modulators of FC and the various data sets used in the study are slightly different on these parameters. One wonders if predictability would improve once such influences are corrected for. It is also likely that age and gender themselves influence on head motion. At least the extent of head motion contribution to predictability needs to be identified, with/without age gender effects.

3) The discovery and validation approach is very good. However, it is all based on data collected in Japan. The results would be more convincing if they can generalize to data collected in other centers. Perhaps the authors can test some of the Drysdale data that is publicly available. This is specially important as the discussion mentions that some of the specific links are shared between them and Drysdale.

4) The parcellation scheme used seems unique to the group. It would be nice if higher/lower resolution parcellations can also be tested. This would address the extent to which observed results are robust and independent of the specifics of parcellation.

5) Head motion remains a main worry in constructing FC matrices. It would be nice to show that resultant matrices were NOT or minimally affected by head motion.

6) Although the authors claim to use independent discovery and validation data sets. It is not always clear that this separation was properly followed. Please indicate more clearly when it was or was not.

7) Obtained results would be further strengthened if identified links or some subset of them actually tracked the primary variable on which the data are being separated: BDI scale. 

Overall, the paper is a very good effort but can be further clarified and needs additional analysis to convince the reader of all the their claims.

---

## [Decision Letter · Decision Letter 2]

7 Oct 2020

Dear Dr Yamashita,

Thank you for submitting your revised Research Article entitled "Generalizable brain network markers of major depressive disorder across multiple imaging sites" for publication in PLOS Biology. I have now obtained advice from the original reviewers and have discussed their comments with the Academic Editor. You will note that reviewer 3, Apkar Vania Apkarian, has revealed his identity. Please accept my apologies for the delay in sending the decision below to you.

Based on the reviews, we will probably accept this manuscript for publication, assuming that you will modify the manuscript to address the remaining points raised by reviewer1. Please also make sure to address the data and other policy-related requests noted at the end of this email.

We expect to receive your revised manuscript within two weeks. Your revisions should address the specific points made by each reviewer. In addition to the remaining revisions and before we will be able to formally accept your manuscript and consider it "in press", we also need to ensure that your article conforms to our guidelines. A member of our team will be in touch shortly with a set of requests. As we can't proceed until these requirements are met, your swift response will help prevent delays to publication.

- a cover letter that should detail your responses to any editorial requests, if applicable

*Copyediting*

*Published Peer Review History*

*Early Version*

Sincerely,

Gabriel Gasque, Ph.D.,

Senior Editor,

ggasque@plos.org,

PLOS Biology

DATA POLICY:

-- Please label panel b in Fig 4.

-- Please check your S1 Data file because you include that for Figure 4c, but there’s no panel c in Figure 4.

-- Please provide data for Figure 5bc. 

-- Please include the X-axis values for data for Fig S1. 

-- Please also ensure that each figure legend in your manuscript includes information on where the underlying data can be found (S1 Data).

Reviewer remarks:

Reviewer #1: The authors have responded very comprehensively to my previous comments on this paper. I would still maintain, however, that distinguishing between MDD and healthy status in clinical practice is straightforward for an experienced clinician. The utility of having a biological marker is to understand underlying pathophysiological mechanisms of illness to guide treatment choice and the future development of novel interventions for a given disorder. The authors should highlight this in the Introduction of the paper.

Reviewer #2: The authors have addressed my comments satisfactorily. 

Reviewer #3: The authors have properly addressed all issues raised by all reviewers. They comprehensively address a long list of questions and incorporate the issues in the introduction and discussion, and properly report all new results either in the main body or in the extensive supplements.

---

## [Editor Report · Decision Letter 3]

2 Nov 2020

Dear Dr Yamashita,

On behalf of my colleagues and the Academic Editor, Tor D. Wager, I am pleased to inform you that we will be delighted to publish your Research Article in PLOS Biology. 

PRODUCTION PROCESS

Before publication you will see the copyedited word document (within 5 business days) and a PDF proof shortly after that. The copyeditor will be in touch shortly before sending you the copyedited Word document. We will make some revisions at copyediting stage to conform to our general style, and for clarification. When you receive this version you should check and revise it very carefully, including figures, tables, references, and supporting information, because corrections at the next stage (proofs) will be strictly limited to (1) errors in author names or affiliations, (2) errors of scientific fact that would cause misunderstandings to readers, and (3) printer's (introduced) errors. Please return the copyedited file within 2 business days in order to ensure timely delivery of the PDF proof. 

If you are likely to be away when either this document or the proof is sent, please ensure we have contact information of a second person, as we will need you to respond quickly at each point. Given the disruptions resulting from the ongoing COVID-19 pandemic, there may be delays in the production process. We apologise in advance for any inconvenience caused and will do our best to minimize impact as far as possible.

EARLY VERSION

PRESS 

Kind regards,

Alice Musson

Publishing Editor, 

PLOS Biology

on behalf of

Gabriel Gasque,

Senior Editor

PLOS Biology